

# Using barometric time series of the IMS infrasound network for a global analysis of thermally induced atmospheric tides

Patrick Hupe[1], Lars Ceranna[1], and Christoph Pilger[1]

[1]BGR, Hanover, 30655, Germany

*Correspondence to*: Patrick Hupe (Patrick.Hupe@bgr.de)

**Abstract.** The International Monitoring System (IMS) has been established to monitor compliance with the Comprehensive Nuclear-Test-Ban Treaty and comprises four technologies, one of which is infrasound. When fully established, the IMS infrasound network consists of 60 sites homogeneously distributed around the globe. Besides its primary purpose of determining explosions in the atmosphere, the recorded data reveal information on other anthropogenic and natural infrasound

sources. Furthermore, the almost continuous multiyear recordings of differential and absolute air pressure allow for analysing the atmospheric conditions. In this paper, spectral analysis tools are applied to derive atmospheric dynamics from barometric time series. Based on the solar atmospheric tides, a methodology for performing geographic and temporal variability analyses is presented which is supposed to serve for upcoming studies related to atmospheric dynamics. The surplus value of using the IMS infrasound network data for such purposes is demonstrated by comparing the findings on the thermal tides with previous

studies and the Modern-Era Retrospective analysis for Research and Applications, version 2 (MERRA-2), which well represents the solar tides in its surface pressure fields. Absolute air pressure recordings reveal geographic characteristics of atmospheric tides related to the solar day and even to the lunar day. We therefore claim the chosen methodology of using the IMS infrasound network to be applicable for global and temporal studies on specific atmospheric dynamics. Given the accuracy and high temporal resolution of the barometric data from the IMS infrasound network, interactions with gravity waves and

planetary waves can be examined in future for refining the knowledge of atmospheric dynamics; e.g., the origin of tidal harmonics up to 9 cycles per day as found in the barometric data sets. Data assimilation in empirical models of solar tides would be a valuable application of the IMS infrasound data.

## 1 Introduction

The International Monitoring System (IMS) was established to monitor compliance with the Comprehensive Nuclear-Test-

Ban Treaty (CTBT) which aims at banning all kinds of nuclear explosions (CTBTO Preparatory Commission, 2017a). The IMS is dedicated to detect explosions down to a threshold of 1 kt TNT equivalent. Three waveform technologies, i.e. seismology, hydroacoustics, and infrasound, cover the entire range of conceivable explosion environments on Earth which are underground, underwater, and atmospheric, respectively.



This study is based on the infrasound network component of the IMS. Besides nuclear monitoring, it also contributes to the project called Atmospheric dynamics Research InfraStructure in Europe (ARISE; Blanc et al., 2017). ARISE aims at designing an infrastructure which combines atmospheric observation technologies, e.g. ground-based wind and temperature measurements (Le Pichon et al., 2015) with infrasound measurements, to retrieve 4D information of the atmosphere. The

objective of ARISE is to enhance weather forecasts and climate models by a more accurate knowledge of atmospheric dynamics in the middle atmosphere (Braathen, 2013; Blanc et al., 2017). The latter issue is of great importance since gravity waves, tidal waves, and planetary waves transport energy and momentum. These waves, which by majority originate in the troposphere, modify the circulation patterns by breaking in the stratosphere and mesosphere (Fritts and Alexander, 2003). Vice versa, there is robust evidence that this variability forced in the middle atmosphere can lead to modifications of tropospheric

circulation systems and therefore affects the weather (Baldwin and Dunkerton, 2001). Moreover, better knowledge of the atmospheric dynamics is essential in the context of the future verification of the CTBT.

With regard to the dynamics and their temporal and spatial variability in the troposphere, this study focuses on absolute air pressure recordings at the IMS infrasound arrays. The potential of infrasound sensors for broad band measurements and the detection of atmospheric dynamics was pointed out by Blanc et al. (2010). Several studies used infrasound measurements for

investigating variations in the middle and upper atmosphere (e.g. Donn and Rind, 1972; Le Pichon et al., 2005; Assink et al., 2012; Smets and Evers, 2014). Here a methodology is described for systematically analysing geographic and temporal variability of atmospheric dynamics' features based on the IMS infrasound network. A similar approach was taken by Marty et al. (2010). We particularly focus on the atmospheric tides and consider up to 13 years of barometric data.

Tides can be traced to two excitation mechanisms (Chapman and Lindzen, 1970). One is gravitational forcing by the Sun and

the Moon. It is commonly known as cause of the periodic rise and fall of the sea level, i.e. the ocean tide, the period of which is related to the lunar day (24 h and 50 min). Since the gravitational pull of the Moon is considerably larger than that of the Sun (e.g. Thomson, 1882; Lindzen and Chapman, 1969), the tidal signature in air pressure data should be related to the lunar day. However, the predominating tidal harmonics found in air pressure recordings are clearly related to the solar day (24 h). Therefore, a second excitation mechanism must exist. It is thermal forcing by the Sun's radiation (Chapman and Lindzen,

1970). Unless otherwise stated, mentioning atmospheric and solar tides throughout this article primarily implies the thermally excited tides with periods related to the solar day. In particular, we focus on the diurnal (24 h), semidiurnal (12 h), and terdiurnal tide (8 h) but we can also identify pressure oscillations with up to 9 cycles per day (cpd) by spectral analyses. The different harmonics result from periodic insolation absorption and latent heat release in various layers of the atmosphere (Forbes and Gillette, 1982). Non-linear wave interactions are considered as another source of the tidal harmonics, in particular those below

a period of 12 h (Moudden and Forbes, 2013).

Moreover, one has to differentiate between migrating and non-migrating components of the aforementioned tides. The former propagate westward, following the apparent motion of the Sun, while the latter do not (Chapman and Lindzen, 1970). When Haurwitz (1965) identified non-migrating components in barometric recordings, he related them to the irregular distribution of land masses. Dai and Wang (1999) stated that the non-migrating tides can reach large amplitudes due to differences of



sensible heat flux on land and sea, respectively. One prominent small-scale example causing a local (non-migrating) diurnal oscillation is the land–sea breeze pattern (Chapman and Lindzen, 1970). In general, the migrating tides prevail over the non-migrating in terms of amplitude (Haurwitz, 1965). Insolation absorption by water vapour and ozone is seen as primary source of the migrating tides (Butler and Small, 1963; Chapman and Lindzen, 1970; Whiteman and Bian, 1996), but cloud effects,

precipitation and latent heating are also considered to be a source of tidal variations, especially for the semidiurnal tide in the tropics (Dai and Wang, 1999). In this study, we will not examine the different sources but rather the characteristics of the various tides detected within a global network.

The infrasound network of the IMS is described in more detail in Sect. 2. Section 3 deals with the selection of the data. Information on the evaluation tools applied is given in Sect. 5. Results on atmospheric tides are presented in Sect. 6. The

discussion included is focused on the potential of the IMS infrasound network for geographic variability analyses in the context of our findings and previous studies on atmospheric tides. Whilst several of those studies addressed the tidal effects and characteristics in the middle (e.g. Forbes, 1984) and upper atmosphere (Forbes and Garrett, 1979; Forbes, 1990; Thayaparan, 1997; Zhao et al., 2005), we concentrate, for reasons of comparability, on troposphere-based observations and theories (e.g. Haurwitz, 1956; Kertz, 1956; Dai and Wang, 1999). We compare our results with the Modern-Era Retrospective analysis for

Research and Applications, version 2 (MERRA-2), which is introduced in Sect. 4.

## 2 The IMS infrasound network

Infrasound is defined as the frequency range between approximately 3.3 mHz, i.e. the acoustic cut-off frequency, and the threshold of human-audible sound, which is approximately 20 Hz. The acoustic cut-off frequency depends, among others, on altitude, temperature, and humidity. The same applies to the speed of sound at which infrasound can travel through the

atmosphere over large distances (Evers and Haak, 2010). This feature is a result of minor attenuation within the abovementioned frequency range. Therefore, infrasound is highly applicable for detecting explosions. Moreover, ducting in the troposphere, in the stratosphere, and in the lower thermosphere contributes to the detection of explosions far away from the source (Drob et al., 2003). In the centuries after the World War II, the infrasound technology was already used to detect significant nuclear explosions in the atmosphere (Christie and Campus, 2010). In the late 1990s, the IMS infrasound network

construction was initiated in terms of CTBT verification purposes to detect and locate even small explosions with a TNT equivalent of 1 kt in the Earth's atmosphere. Its 60 stations are therefore more uniformly distributed than previous, smaller networks (Blanc et al., 2010). As of the end of November 2017, 49 stations (see Fig. 1) have been certified by the CTBTO and are in permanent operation. Another three stations are being under construction and eight sites are still in the process of planning (CTBTO Preparatory Commission, 2017b). Meanwhile, the detection capability of the network was demonstrated

based on natural sources of infrasound, e.g. the Chelyabinsk meteorite over Russia in 2013 (Le Pichon et al., 2013; Pilger et al., 2015).





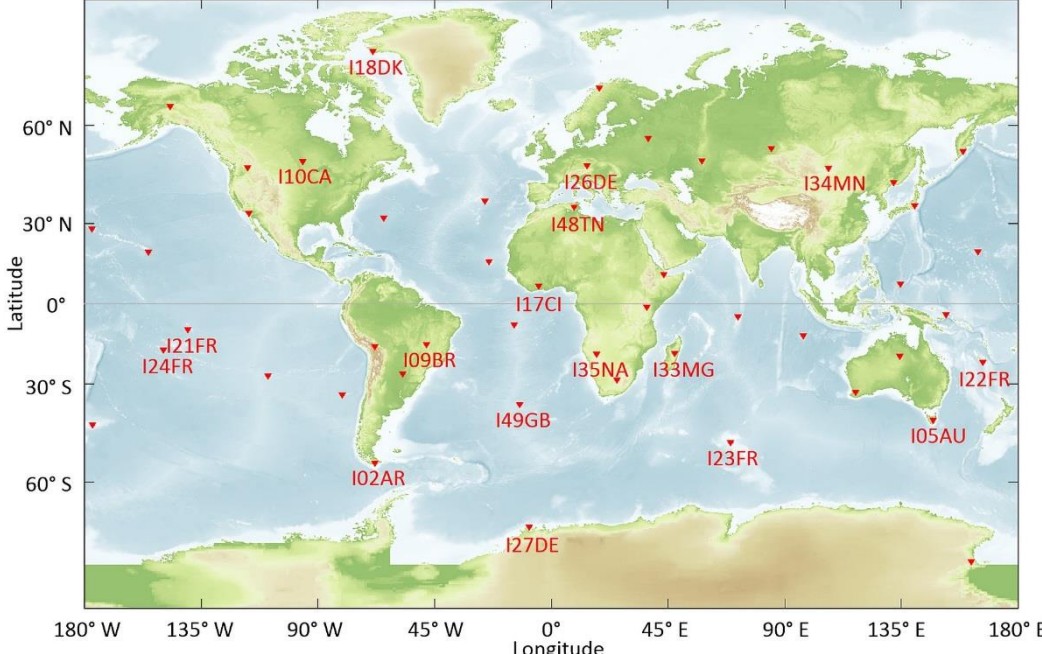

**Figure 1: Station map of the IMS infrasound network as part of the CTBT verification regime. Each red triangle represents a certified array (November 2017). The stations labelled are the data base for the study which is subject of this paper.**

Each IMS infrasound station is constructed as an array consisting of at least four sensors, i.e. microbarometers with a sensitivity down to 1 mPa. To enhance a station's detection capability in a noisy environment, arrays are equipped with acoustic filters providing additional noise reduction at each sensor (Hedlin et al., 2003; Alcoverro and Le Pichon, 2005). Altogether, each array serves as acoustic antenna that gives indication of azimuth direction and apparent wave velocity of a passing signal. In addition to differential pressure, MB2000 and MB2005 microbarometer sensors also record absolute air pressure (Ponceau and Bosca, 2010). This channel is included in the microbarometer with a bandwidth of DC to 40 Hz, whereas the differential pressure channel operates at a bandwidth of 0.01 Hz to 27 Hz (MARTEC, 2006). Sensor electronic noise amounts to 2 mPa ($2 * 10^{-5}$ hPa) which is negligibly small in terms of atmospheric dynamics relevant in this study.

## 3 Data selection and handling

As previously mentioned, the focus in this study is on geographic variability of atmospheric tides based on absolute pressure recordings. From the 49 infrasound stations certified so far, we have selected 17 sites as data base (labelled triangles in Fig. 1). The selection accounted for various aspects, one of which is data availability. Most of the microbarometers record absolute pressure but the time series cover non-uniform periods. This results from subsequent installation and certification of new sites since the CTBTO initiated the design of the IMS. When selecting the stations, we have also aimed at a uniform geographic distribution of the stations with limited periods of missing data.





The sampling rate of absolute air pressure data is generally 1 Hz. At some stations, recordings of even 20 Hz are available. Taking into account that significant air pressure changes take several minutes to hours rather than seconds, we have chosen a time step of 1 min and have therefore reduced the enormous amount of data. As a consequence of very different topographic locations of the infrasound arrays all over the world, e.g. I27DE in Antarctica and I17CI in Ivory Coast, the barometric time

series naturally differ from each other. For reasons of comparability and due to the fact that the recorded air pressure is not reduced to mean sea level, we concentrate on air pressure fluctuations around the annual mean. For this purpose, the respective annual mean is subtracted from the annual data sets, resulting in fluctuations around zero.

In case of multiple recordings per array the aforementioned procedure is adapted to each element's time series. Afterwards, as schematically illustrated in Fig. 2, the median of the mean-free recordings is extracted. Overall, we have only one record per

station and are able to diminish the problem of temporary missing data at a single sensor.

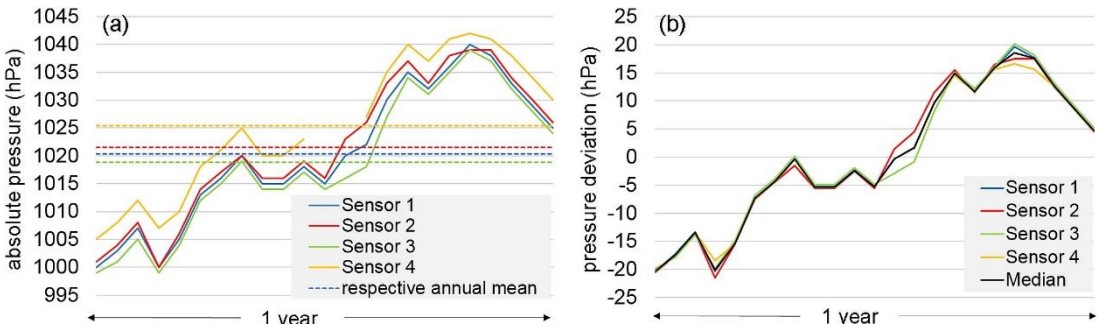

**Figure 2: Schematic illustration for handling an annual data set of a four-sensor array. Different pressure at the various sensors (a) can result from the calibration or from different altitudes since the arrays have an aperture of 1-3 km (Evers and Haak 2010). Here, temporarily missing data for sensor 4 are without significant consequence when deriving the median (b) as time series for a multi-**
**sensor station.**

Nonetheless, the majority of stations only provides one data set of absolute air pressure, corresponding to one sensor. As a consequence, supplementing the occurring errors or missing data is beyond the realms of possibility. Therefore, obvious erroneous values are handled as missing data; i.e., values which deviate from the mean by more than 4 times the standard deviation. Such outliers are additionally examined on plausibility (e.g. cyclones). Figure 3 provides an overview on data

availability of absolute pressure for the selected stations. The lack of data is related to, among others, station maintenance, power failures by various reasons, and erroneous data.

The remaining data sets are up to 13 years long and thus sufficient for being analysed with regard to dynamic features since long period phenomena such as planetary waves as well as short period phenomena such as gravity waves and tides are well represented within this time interval. However, the time interval is likely too short to identify reliable trends, e.g. associated

with climate change. Since our analyses are based on the pressure deviation from the respective annual means, long-term trends are removed anyway. The remaining time series clearly reveal characteristics of their geographic location. As an example, two data sets are shown in Fig. 4, namely those of the midlatitudinal station IS26 (Germany, Fig. 4a) and the tropical station IS21 (Marquesas Islands, French Polynesia, Fig. 4b).





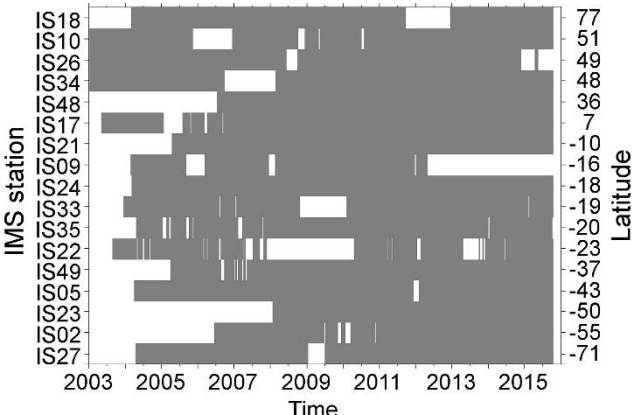

**Figure 3: Daily based availability of absolute pressure data at the selected IMS stations. White boxes indicate missing data.**

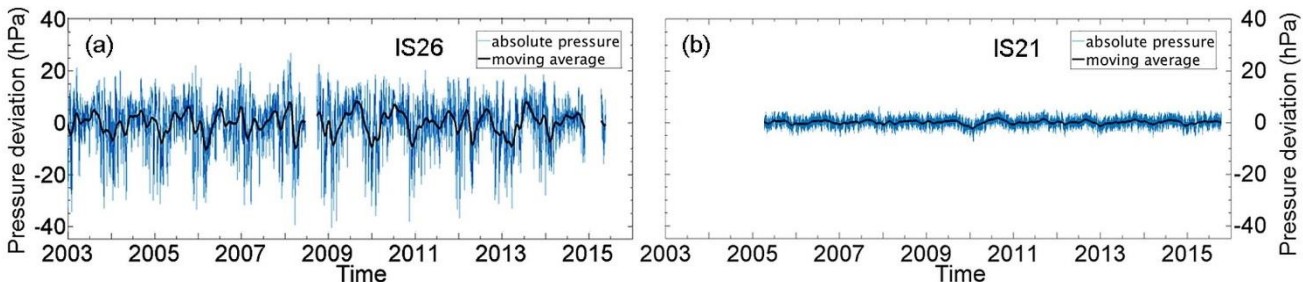

**Figure 4: Absolute pressure data recordings of the IMS infrasound stations IS26 (a) and IS21 (b) as deviations from the annual**
**means. The moving average highlights the superordinate annual variation.**

In the tropics at IS21, the annual cycle is characterized by comparatively small amplitude. The absence of large-scale pressure gradients leads to a dominant signature of small-scale phenomena such as thunderstorms, or small-amplitude phenomena such as the solar tides. In the tropics, the semidiurnal tide's amplitude can amount to 1.3 hPa (Dai and Wang, 1999; Hoinka, 2007; Schindelegger and Ray, 2014). Even though this tidal amplitude is only about 0.1 % of the absolute air pressure, its proportion
of the annual pressure fluctuation amplitude amounts to about 20 %.

At IS26, however, the annual amplitude (about 30 hPa) masks small-scale amplitudes like those of the atmospheric tides. The negative deviations which predominantly occur in winter are larger than the positive deviations throughout a year. In winter, temperature gradients between equator and North Pole are largest due to differing solar radiation and, thus, energy balance. The compensating processes result in stronger westerlies and more intense (low pressure) cyclones than in summer.

**4 The MERRA-2 reanalysis data**

MERRA-2 is NASA's latest atmospheric reanalysis of the modern satellite era provided by the Global Modelling and Assimilation Office (Gelaro et al., 2017). Data are available on a 0.625° $x$ 0.5° grid, beginning in 1980 (Bosilovich et al., 2016). Since ECMWF's ERA-interim provides reanalysis data at the meteorological main times (0, 6, 12, 18 UTC) only,





interpolation methods would be necessary for detailed studies of tides with periods shorter than 24 h (Ray and Ponte, 2013). MERRA-2 has the benefit of providing data at an interval of 3 h either as instantaneous or as time-averaged fields (Bosilovich et al., 2016). For this study we retrieved the 3 h time-averaged surface pressure field (GMAO, 2017). Data have been interpolated at the locations of the 17 IMS stations defined in Sect. 3. Analogue to Fig. 4, MERRA-2 time series for IS26 and
IS21 are shown in Fig. A1.

To analyse the tidal effects at various IMS infrasound stations and to derive other features related to atmospheric dynamics from the pressure fluctuations, e.g. planetary waves, we apply various spectral analysis methods on both the IMS barometric data and the MERRA-2 data.

## 5 Spectral analysis tools

Fourier methods' outputs comprise, inter alia, wavelets and the power spectral density (PSD) of the time series. Computing the PSD of a time series helps to get a first impression of the data set. In contrast to Marty et al. (2010) who applied the PSD on intervals of 12 d, we have chosen time windows of 6 months. Consequently, single days missing do not significantly affect the analysis. When considerable data gaps exist in the time series (cf. Fig. 3), the PSD calculation is rejected for the particular year. An example of a PSD is given in Fig. 5a, based on the pressure fluctuation at IS26 from 2003 to 2015. A combined PSD
plot for all stations selected is provided in Sect. 6. Additionally, the PSD is calculated based on the Lomb–Scargle algorithm which is capable of handling missing data without reducing the data basis (Lomb, 1976; Scargle, 1982). An example is given in Fig. 5b. A confidence level is added to the Lomb–Scargle PSD spectrum for classifying the peaks' significance. The Lomb–Scargle PSD of the MERRA-2 time series is in good agreement.

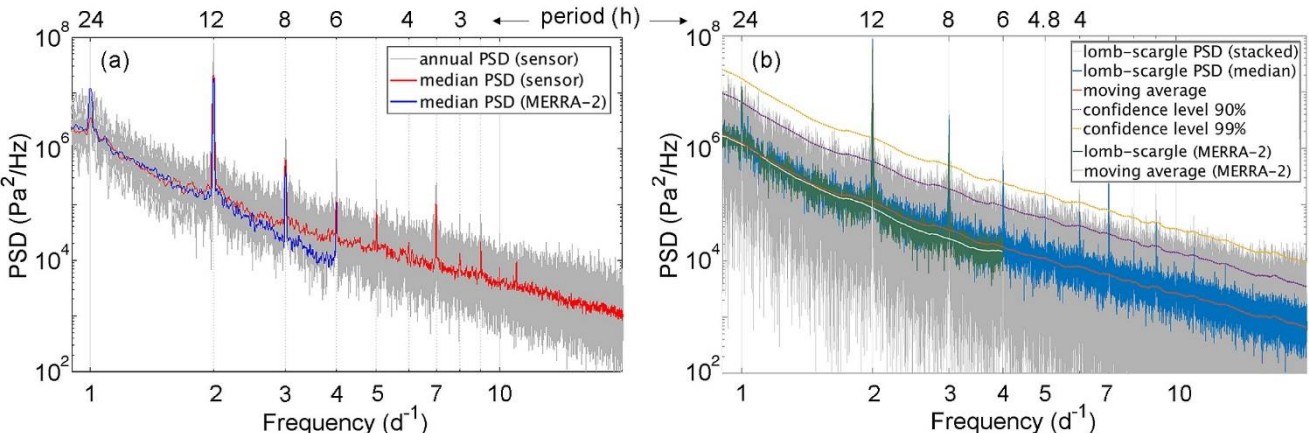

**Figure 5: PSD plots for IS26. (a) The red curve depicts the median PSD for the entire period from 2003 to 2015 as derived from the barometric sensor. Each of the grey PSD curves belongs to a single year within this period. Panel b shows the Lomb–Scargle PSD spectrum based on the entire time series. In both panels, the solar tides appear as sharp peaks. The MERRA-2 reanalysis reveals a higher PSD for the diurnal tide.**



Another useful and often used tool for geophysical studies is to compute a wavelet analysis (Daubechies, 1992). In our study it is based on the Morlet wavelet function. The barometric time series can be analysed in the time–frequency space, enabling the determination of both the dominant frequencies and their temporal variation (Torrence and Campo, 1998). Figure 6 includes the wavelet power spectrum for station IS26. The semidiurnal tide is highlighted by a bright horizontal line. Since the diurnal

tide is not very prominent at IS26 (cf. Fig. 5), it is hidden in synoptic pressure fluctuations rather than being represented by a distinct line. The semidiurnal tide is however recognizable by a distinct line. The wavelet power spectrum does not provide detailed information on smaller period scales or even seasonal variations within the diurnal period range.

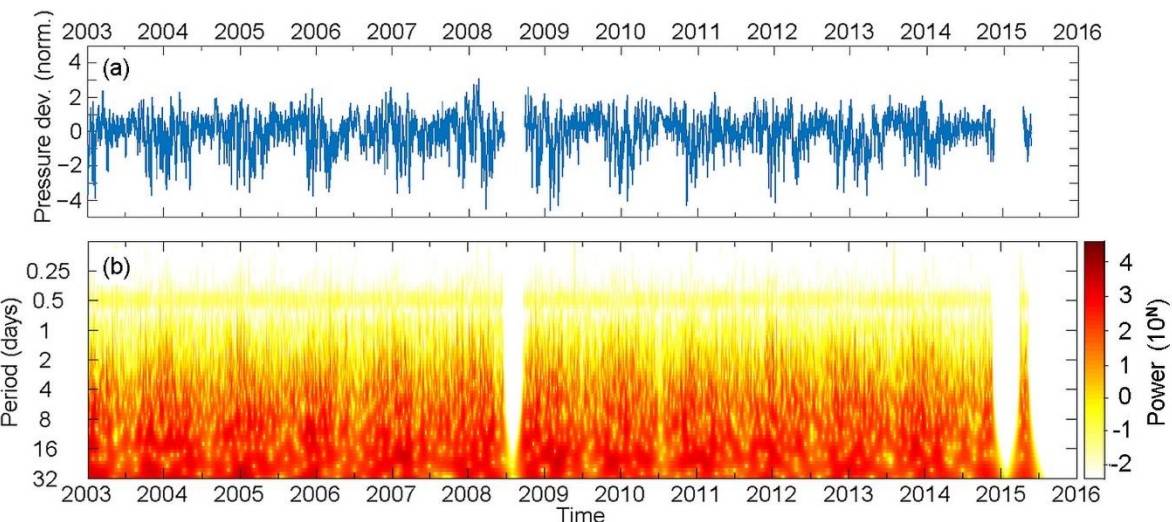

**Figure 6: Wavelet analysis for station IS26 from 2003 to 2015. The time series in panel a corresponds to Fig. 4a but is normalized by**
**its standard deviation which amounts to 8.78 hPa. The resulting power spectrum (b) is colour-coded. The ordinate axis represents**
**the Fourier period in days.**

## 6 Results and discussion

In this section, we highlight and discuss the findings on several dynamic features of the barometric data considered. The focus is on geographic and temporal variability of the solar atmospheric tides.

### 6.1 Geographic variability of the dynamic features

The PSD curves of the selected data sets reveal both consistent and differing features like those shown for IS26 (Fig. 5a). For reasons of comparability the median PSD curves are colour-coded and sorted according to the stations' latitude in Fig. 7. For MERRA-2 data this PSD spectrum is given in Fig. A2.





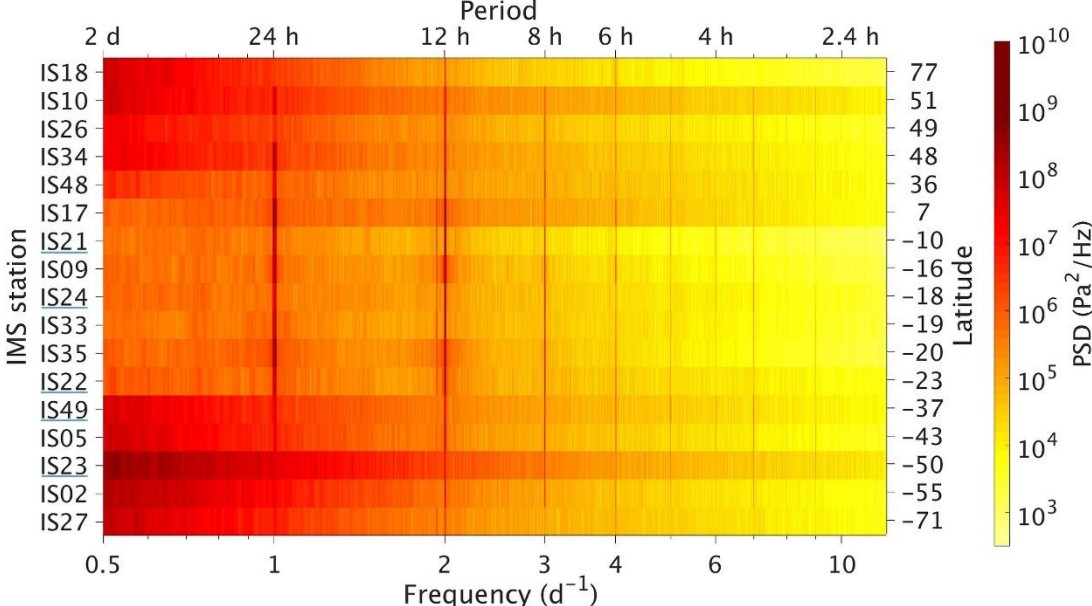

**Figure 7: Median PSD of the selected data sets. The IMS infrasound stations are sorted from north (top) to south (bottom) with the latitudes indicated on the right axis. Stations on (small) islands are underlined. The darker the colour the higher is the PSD. The corresponding spectrum for MERRA-2 data can be found in Fig. A2.**

### 6.1.1 Phenomena in the period scale exceeding 1 d

In general, the PSD increases with decreasing frequency. At periods exceeding 1 d, tropical stations can be clearly distinguished as the PSD values are significantly lower than at latitudes beyond ±30°. The difference is in the order of one power of 10. In the midlatitudes and high-latitudes extratropical cyclones and planetary waves lead to larger pressure fluctuations compared to the tropics, where air pressure fluctuations are relatively small throughout a year as was shown in Fig. 4b. In Fig. 6, the wavelet spectrum provides extended information on planetary wave scales (periods up to 32 d). These are strongest in the winter months when temperature gradients between equator and North Pole are largest.

### 6.1.2 The solar tides

Geographic differences between the tidal effects on air pressure may be worth to discuss in the context of previous studies on atmospheric tides derived from barometric data. In Fig. 7, tidal peaks clearly appear with up to 9 cpd. The most striking peak, in terms of a constantly high PSD, is the semidiurnal tide (12 h). Also 3 cpd and 4 cpd, which are called terdiurnal (8 h) and quarterdiurnal tide (6 h), respectively, can be recognized at the majority of stations. Generally speaking, several tidal modes are dominant throughout the entire hemispheres, starting with a period of 24 h which is known as the solar diurnal tide. According to findings on global pressure oscillations by Haurwitz (1965), whose study was based on more than 200 stations irregularly distributed, the diurnal tide's amplitude decreases with increasing latitude, following the latitudinal decrease of solar insolation. By using the IMS infrasound stations and by considering the picture provided in Fig. 7, we come to a similar





conclusion: The PSD and thus the amplitudes are maximum in the low-latitudes and almost disappear at the high-latitude stations IS18 and IS27. The PSD, however, does not steadily decrease towards the polar regions. For example, IS34 and IS10 are characterized by relatively large PSD values which can be explained by the continental location of the sensors: in summer, the rapid warming of the landmasses causes a stronger diurnal effect. And even within the tropics the PSD values differ

markedly since the amplitude of the diurnal tide strongly depends on local surface conditions (Haurwitz, 1965); e.g. over the ocean it is weaker than over landmasses (Dai and Wang, 1999). For example, the PSD is comparatively weak at maritime stations like IS21, IS24, and IS22. The midlatitudinal stations in the Southern Hemisphere are by majority located on islands or otherwise close to an ocean. Hence, the corresponding PSD values are generally weaker than in the Northern Hemisphere. Dai and Wang (1999) also emphasized a strong diurnal tide over high terrain which might contribute to the prominent PSD

signal at IS34 in Mongolia as well. Extracting the tidal components from the PSD spectrum shows that the diurnal oscillation is not the strongest tide. Haurwitz (1965) even claimed that the amplitude of the semidiurnal tide was "generally considerably larger than that of the diurnal oscillation". Regarding tropical stations, this agrees with Fig. 8, in which the means of monthly amplitudes of the tidal components are given for each station. The amplitude means of the semidiurnal tide are much larger than the diurnal ones for most stations, in particular at the low latitude stations; hence, the 12 h oscillation can often be easily

recognized in barometric recordings at tropical stations (e.g. Oberheide et al., 2015) where large-scale pressure oscillations do not exist (see Fig. 4b and Fig. A3). The pressure maxima occur at approximately 10:00 and 22:00 local solar time (Haurwitz, 1956; Dai and Wang, 1999; Marty et al., 2010) or, in supposedly more precise words, "about 2.3 h before solar noon and solar midnight" (Whiteman and Bian, 1996).

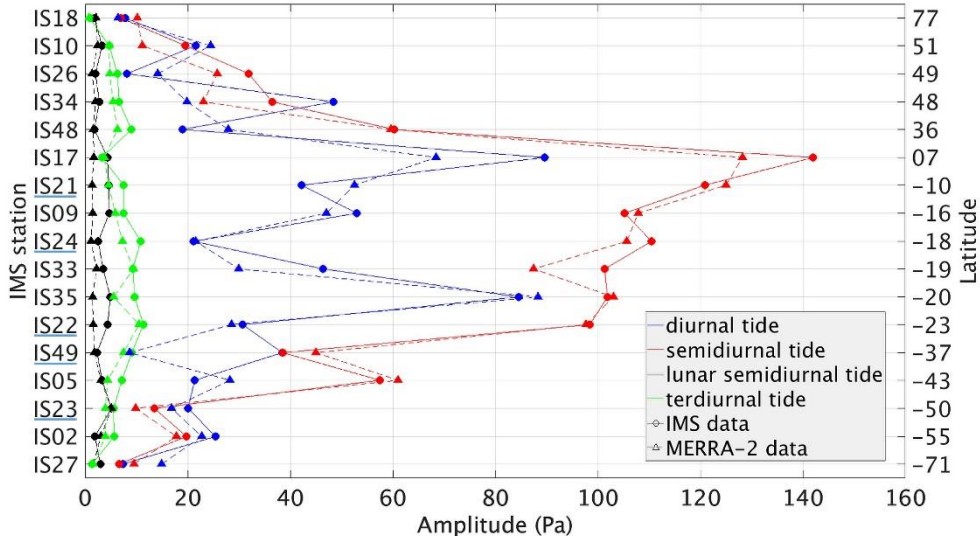

**Figure 8: Comparison of the mean tidal amplitudes as calculated from the IMS sensors' time series and MERRA-2 reanalysis. The**
**mean amplitudes of the solar tides are in good agreement. The increased amplitude of the lunar tide at IS23 can be seen as**
**misrepresentation since it matches the MERRA-2 data in which the lunar tides are not represented (Fig. A2) as in Fig. 7.**





At high latitude stations, however, the diurnal tide's amplitude is larger than the semidiurnal component. The semidiurnal tide is driven, amongst other origins, by the absorption of solar radiation by water vapour (Whiteman and Bian, 1996). Both the concentration of water vapour and the insolation are highest in the tropics; hence, the maxima can consequently be found in the low latitudes where the effect of the diurnal tide is exceeded; e.g. the amplitude mean found for IS17 amounts to 1.4 hPa

(diurnal tide: 1 hPa).

The amplitude means of the terdiurnal tide are less by about one power of 10 at tropical stations. Amplitude maxima can be found in the tropics (IS22: 13 Pa) and minima at high latitudes where the amplitude is less than 2 Pa. Spectral analyses of MERRA-2 time series resulted in slightly lower estimates of the terdiurnal tide. These can be traced back to the 3 h sampling of reanalysis data which might be too sparse for exact estimates of the 8 h cycle.

The generally good agreement with amplitude means calculated from MERRA-2 time series underlines the IMS infrasound network's capability for studying geographical variations of atmospheric dynamics such as solar tides or gravity waves. Vice versa, our findings imply that the solar tides' representation in MERRA-2 is accurate.

### 6.1.3 The lunar tides

The most striking PSD peaks in Fig. 7 are clearly related to the solar day. Besides, the individual PSD curves of some IMS

stations reveal a secondary semidiurnal peak. It can also be recognized in Fig. 7 for a couple of equatorial stations but not in the equivalent MERRA-2 analysis (Fig. A2). The period is 12 h and 25 min and thus exactly half a *lunar* day. It is striking that at IS27 the lunar tide seems even more prominent than the diurnal solar tide. Other significant peaks are generally found at tropical stations such as IS17 and IS21. Figure 9 shows the Lomb–Scargle PSD spectra of IS27 and IS21, both showing a highly significant peak next to the main (solar) semidiurnal peak. The peak is not matched by MERRA-2. The monthly mean

amplitude of the semidiurnal oscillation related to the lunar day given in Fig. 8 is very small in relation to the solar tides.

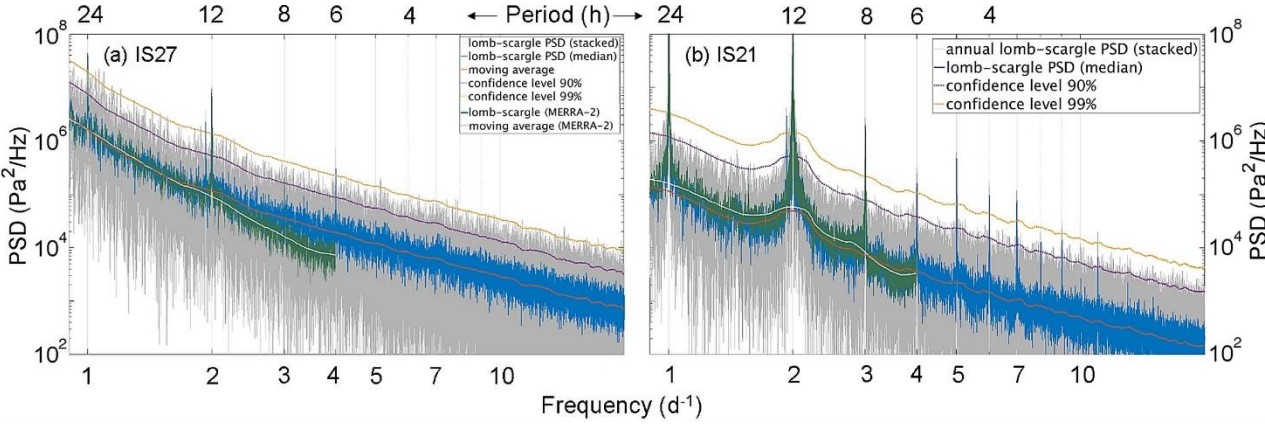

**Figure 9: Lomb–Scargle PSD curves for (a) IS27 (Antarctica) and (b) IS21 (Marquesas Islands). The peaks closely left of the predominant semidiurnal peak are related to the lunar day. This peak is not present in MERRA-2 reanalysis.**




A study of Chapman and Westfold (1956) indicated that the global distribution of the lunar semidiurnal tide is similar to that of the solar counterpart. Hence, the mean annual amplitude is maximum in equatorial regions and decreases with increasing latitude (Haurwitz and Cowley, 1969; Schindelegger and Dobslaw, 2016). Combined with its generally small amplitude (Lindzen and Chapman, 1969), this explains why distinct signatures of the lunar semidiurnal tide (Fig. 7) are mainly concentrated on tropical stations in our study. In this context, the evidence of a tidal peak related to the lunar day at the Antarctic station IS27 is even more surprising. However, the monthly amplitude mean we found at IS27 is only 3 Pa. For MERRA-2 data we calculated a mean amplitude of 2 Pa, although the lunar tide seems not well represented (see Figs. 9a and A2). The significant PSD peak still raises the question about the excitation mechanism of the tidal signal since the dynamic effect of the gravitational pull of the Moon seems to be too low in the Antarctic region. As IS27 is mounted on the Ekstroem Ice Shelf, the ocean tide may affect barometric recordings to a small but considerable extent by vertically lifting the sensor. Vertical sensor lifting can be induced by the body tide, the ocean tide itself, and ocean tide loading effects. Near IS27, these excitations amount to about 4 Pa (Schindelegger and Dobslaw, 2016). The open-ocean tide amounts to 0.5 m (Kohyama and Wallace, 2014). To a certain extent, the Ekstroem Ice Shelf moves with the ocean tide which may enhance the barometric effect of the lunar tide as the vertical movement translates into a barometric signal.

## 6.2 Seasonal variability of the solar tides

Focusing on the solar tides again, the monthly PSD is computed to derive the seasonal variability of the diurnal, semidiurnal and terdiurnal tide. For the period range between 3 h and 5 d the monthly PSD is shown for station IS21 in Fig. 10a. Several tidal harmonics are highlighted by distinct horizontal lines. At first sight, the semidiurnal oscillation is widely of continuous power. To specify the seasonal characteristics, the variances of the most dominant tides are given in Fig. 10b. It turns out that the semidiurnal tide's power is not that uniform throughout a year but mostly exhibits two maxima. They predominantly occur in the equinoctial months, which is in line with studies by Hann (1918) and by Haurwitz and Cowley (1973). The diurnal tide's variance is generally smaller than the semidiurnal tide's variance. The maxima seem to occur without a clear periodic cycle. The terdiurnal tide maximizes twice per year. Besides its absolute annual maximum in winter, a secondary maximum with less power can be detected in summer. Ray and Poulose (2005) evidenced this seasonal variation of the terdiurnal tide for barometric recordings in the United States. We can now validate it for the IMS infrasound stations distributed all over the world. The variation of the monthly mean amplitudes (not shown) of both MERRA-2 and IMS data is in line with this annual cycle.

In Fig. 11, the occurrence months of the absolute variance's maxima between 2003 and 2015 are depicted (one maximum each year). A month's rectangle is highlighted in the colour of the respective tidal harmonic if in at least three years (i.e. approximately 25 %) the variance's annual maximum was detected within this month. This enables to detect seasonal patterns for each of the tides.

The annual maximum of the terdiurnal tide (green) is on average detected during the winter months. A second, smaller maximum can be detected in the summer months as it has been validated for IS21 in Fig. 10 (Ray and Poulouse, 2005). An



exception is given by the equatorial station IS17 at which the terdiurnal tide's power is relatively constant throughout a year. Besides a small maximum in December or January, the occurrence of another small maximum ranges between March and July, depending on the year considered.

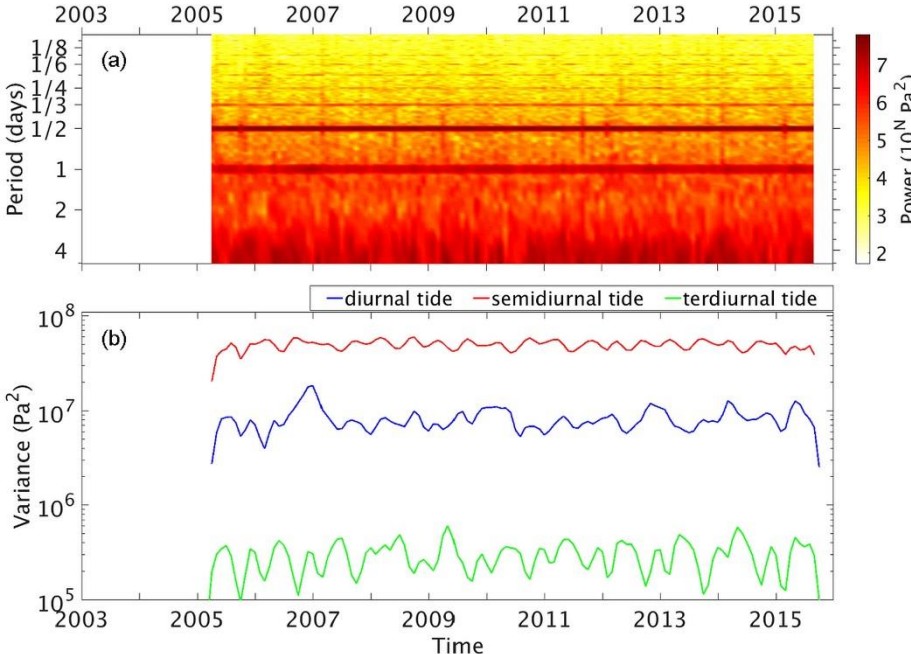

**Fig. 10: Colour-coded PSD spectrum (a) of the monthly air pressure fluctuations at IS21 as function of time and Fourier period, and the variances of the diurnal, semidiurnal, and terdiurnal tide (b).**

The annual cycle of the semidiurnal tide (red) exhibits two maxima during the equinoxes with only a small difference in power (Hann 1918; Haurwitz and Cowley, 1973). The absolute maximum predominantly occurs during the spring equinox. The semiannual cycle showing maxima around the equinoxes was also found in ECMWF reanalysis after interpolation of the 6 h fields (Van den Dool et al., 1997; Díaz-Argandona et al., 2016). Exceptions are the high-latitudinal stations: At IS18 the maxima are detected in northern winter, whereas at IS27 the maxima of the semidiurnal tide are distributed over several months. Because of the low tidal amplitude in high latitudes, variances in the tidal period ranges are certainly affected by noise. As one can also recognize from Fig. 11, the annual variation of the diurnal tide (blue) is not indicated by such a clear pattern like the aforementioned tides. The maxima would be expected in summer when solar radiation leads to increased heating of land masses (Haurwitz and Cowley, 1973; Dai and Wang, 1999). In the midlatitudes also a semiannual cycle is known from observations (Hann, 1918), but the absolute maximum was primarily found in (early) summer. This is also a finding in our observations which agrees with the MERRA-2 time series analysed. However, the annual cycle of the diurnal tide strongly depends on the geographic location of the station. At maritime stations, e.g. those on oceanic islands such as IS21 (Marquesas Islands) or IS22 (New Caledonia), the annual variation of the diurnal tide is small, resulting in a varying appearance of the annual maximum or even no explicit one. Contrarily, at IS34 in Mongolia the maxima of the diurnal tide are always in the





summer season when the Asian continent is dominated by warm air masses. In winter, when the high-pressure system with the cold air mass prevails, the diurnal tide is less powerful. At other stations in the midlatitudes and high-latitudes, the low-amplitude diurnal tide is often hidden in the noise or masked by synoptic pressure gradients. In particular, this regards the stations IS02, IS10, IS18, IS23, IS26, and IS27. Therefore, the seasonal variation found here is not that clear as it would be

expected from the theory of excitation by insolation (e.g. Dai and Wang, 1999). Instead of an annual cycle exhibiting one maximum in summer (Haurwitz and Cowley, 1973), maxima can partly be found in winter (Fig. 11) since the largest pressure gradients are present in those months.

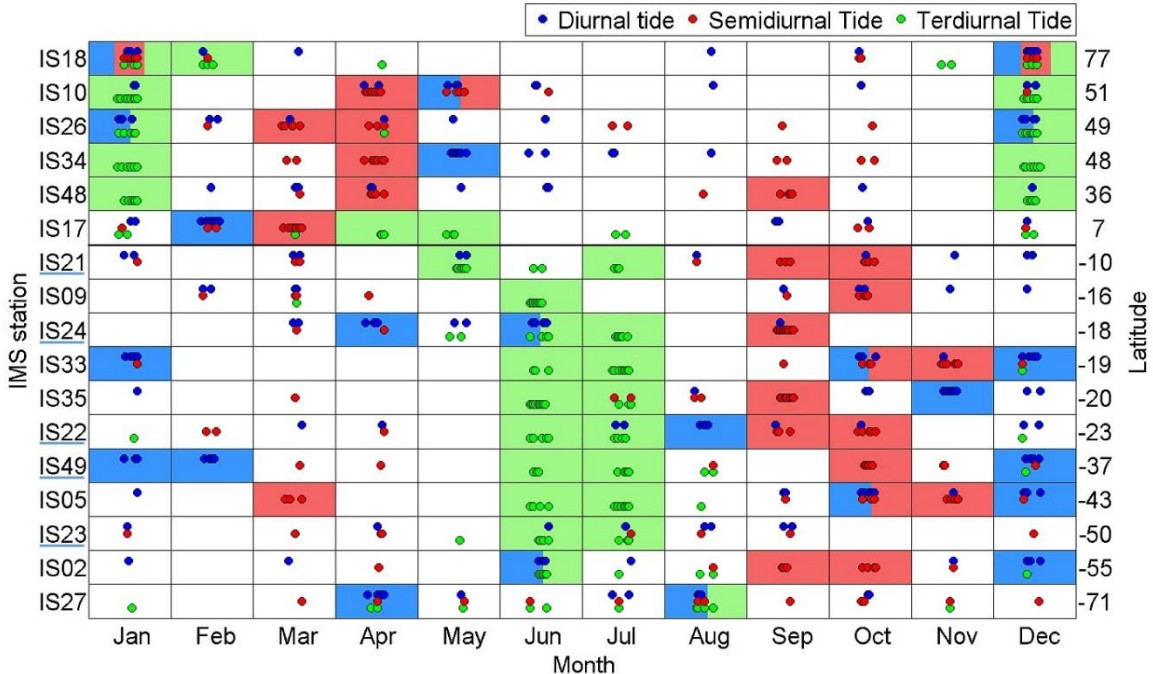

**Fig. 11: Detection times of the primary annual maxima of the tides' variances between 2003 and 2015. The thick, black line depicts**
**the transition from northern to southern latitudes. For the IMS infrasound station indicated on the left axis, each dot marks the month in which the absolute maximum of each year between 2003 and 2015 appeared. The offsets between the dots serve for the purpose of distinguishability. By highlighting accumulations of annual maxima, seasonal patterns and its differences between the Northern Hemisphere and the Southern Hemisphere become apparent.**

## 7 Conclusions

Based on absolute pressure data of the global IMS infrasound network we have compared signatures of the atmospheric tides, derived by using spectral analysis tools, with surface pressure reanalysis from MERRA-2. We intended to show the capability of this network for future applications on temporal and geographic variability studies of atmospheric dynamics and assumed to focus on the thermally forced tides by the Sun. These generally appear much stronger in atmospheric recordings than gravitationally induced tides (Chapman and Lindzen, 1970). The sensors of the IMS infrasound network are also capable of

representing the gravitationally forced lunar tide.



We have detected the lunar semidiurnal tide at almost all IMS stations despite its small amplitude (Lindzen and Chapman, 1969). Its PSD peak is highly significant at all tropical stations. The distinctness of this signature in the low-latitudes is certainly also a result of the accurate sensitivity of barometric sensors at the IMS infrasound stations. Following the review on atmospheric tides by Lindzen and Chapman (1969), the availability of hourly and even shorter term data also proves as a large

benefit. This is underlined by the fact that the existence of the lunar tide could not be proved in MERRA-2 time series. In contrast, we have even found a significant lunar semidiurnal tide at the Antarctic IMS station, thus far beyond the tropics. We have pointed out that it can be of spurious nature as indirect sensor lifting effect of the ocean tide (Schindelegger and Dobslaw, 2016).

Our spectral analyses have shown that quantitative measures of the diurnal, semidiurnal, and terdiurnal tides, e.g. PSD and

amplitude, are well represented by both IMS recordings and MERRA-2 data. Despite the different sampling intervals of 1 s (IMS) and 3 h (MERRA-2), only slight differences were found which can be related to the spatial resolution of the reanalysis field and local effects at the IMS stations. Our global observation regarding the diurnal and semidiurnal solar tide is widely compliant with previous studies which were based on observations and model analysis. The IMS infrasound network is smaller than previous networks (e.g. Haurwitz, 1965; Dai and Wang, 1999) but it is equipped with highly accurate barometric sensors

which are uniformly distributed over the globe. In future, gravity waves which are in the period range within 1 d could be examined given the accuracy and high temporal resolution of the barometric data from the IMS infrasound network.

Data assimilation in existing empirical models of both solar (e.g. Dai and Wang, 1999; Schindelegger and Ray, 2014) and lunar tides (e.g. Kohyama and Wallace, 2014; Schindelegger and Dobslaw, 2016) at ground level would be a valuable application of the IMS infrasound data. By now, the focus of such empirical models is often limited to the diurnal and the

semidiurnal tide. The accurate barometric IMS data sets also allow for analysing the global characteristics of higher tidal harmonics. For this purpose, the set of selected IMS infrasound stations is still extendable. So far, the 13 years of data availability have revealed valuable findings on the seasonal variation of the terdiurnal tide as well. Other tidal harmonics and their excitation sources could be studied since up to 9 cpd clearly appear in the PSD analysis of our data set. For example, the quarterdiurnal tide has been investigated in few studies only. The one by Pramanik (1926) was just based on continental

stations. By using the IMS infrasound network also maritime regions can be represented by using stations located on islands. We have studied the terdiurnal tide with a period of 8 h by using barometric recordings. To our knowledge, this has only been performed on a more regional scale so far, e.g. in the United States (Ray and Poulose, 2005), or as a collection of data in absence of a uniform network (Hann, 1918). On a global scale, we have clearly identified a semiannual cycle of the terdiurnal tide with maxima in summer and winter. The excitation mechanism of the terdiurnal tide must be different or more diverse

compared to the diurnal and semidiurnal tide, respectively. Wave–wave interactions are considered to be at least a secondary source of atmospheric tides (e.g. Forbes and Wu, 2006; Moudden and Forbes, 2013). As a next step, using the IMS infrasound network for temporal correlation between the tidal harmonics and other dynamics such as planetary waves or gravity waves could lead to clarification regarding such interactions. This is significant in the context of ARISE2 and for other communities with the objective of achieving an enhanced understanding of dynamic processes in the atmosphere; e.g., in the context of the

 

future verification of the CTBT, improved atmospheric models are needed for assessing the IMS infrasound network performance in higher resolution, leading to advanced source characterisation and location.

**Appendix A**

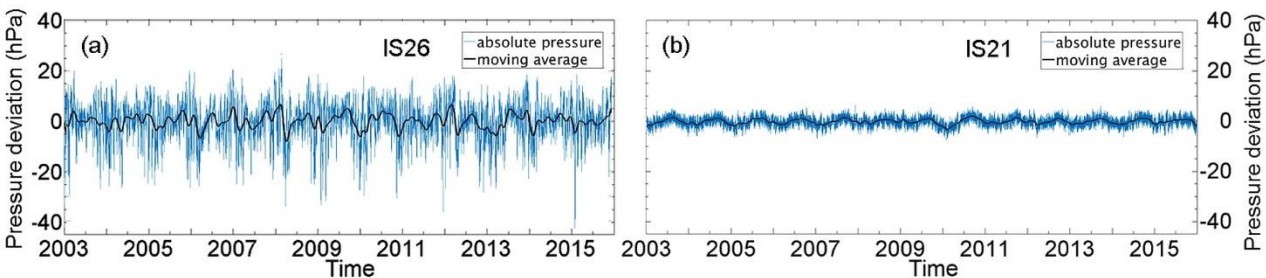

5  **Figure A1: Absolute surface pressure data of MERRA-2 at IMS stations IS26 (a) and IS21 (b) as deviations from the annual means. The moving average highlights the superordinate annual variation. See also Fig. 4.**

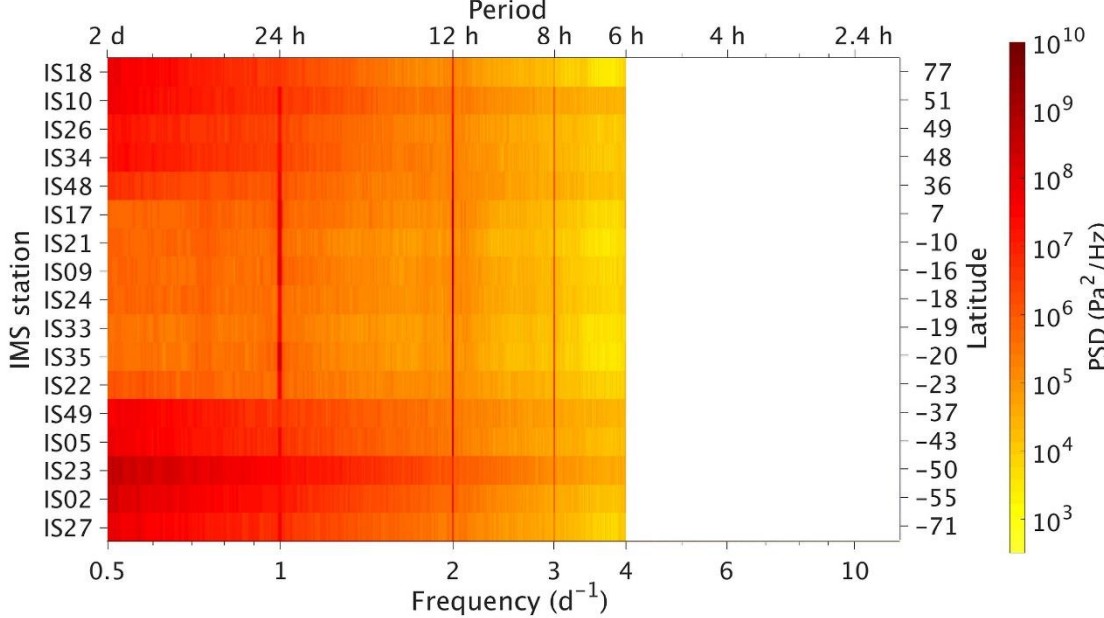

**Figure A2: Median PSD spectrum of the MERRA-2 time series (analogue to Fig. 7). The period range is limited at 6 h due to the sampling interval of MERRA-2 data (3 h). The colour bar is slightly different from Fig. 7. As most important difference, the lunar**
10  **semidiurnal tide is not represented in MERRA-2 data.**





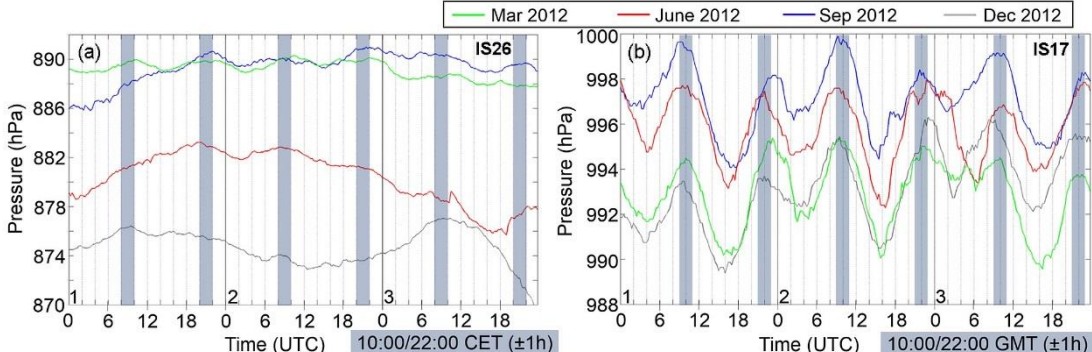

**Figure A3: Absolute pressure on three days of different seasons at (a) IS26 (UTC+01) and (b) IS17 (UTC+00). The semidiurnal tide can clearly be detected by eye-inspection at the tropical station IS17, whereas at IS26 synoptic pressure variations mask the reduced tidal effect in the mid-latitudes. The grey columns indicate the expected times of the semidiurnal tides' pressure maxima (e.g.**
**Haurwitz, 1956). Note that these barometric recordings are exemplarily shown for 2012 and are not necessarily representative for any of the four seasons.**

### Data availability

MERRA-2 data can be accessed online through the NASA Goddard Earth Sciences Data Information Services Centre (GES DISC). The surface pressure reanalysis data used in this study can be retrieved from the Assimilated Meteorological Fields
(GMAO, 2017). Access to the IMS network's data such as barometric recordings of the infrasound stations can be provided by the respective National Data Centres of the CTBTO.

### Author contribution

PH prepared the manuscript and performed the data analyses with support from LC and CP. LC supervised the project. All authors discussed the results and thereby contributed to the final version of the manuscript. The authors declare that they have
no conflict of interest.

### Acknowledgments

The research leading to these results has been performed within the ARISE2 project (http://ARISE-project.eu) at BGR (Federal Institute for Geosciences and Natural Resources) as the German National Data Centre of the CTBTO. The ARISE2 project received funding from the European Commission's Horizon 2020 program under grant agreement 653980. The authors thank
BGR colleague Ole Ross for his technical assistance with MERRA-2 data retrieval. The views expressed herein are those of the authors and do not necessarily reflect the views of the CTBTO Preparatory Commission.



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
