# Peer review of "Using barometric time series of the IMS infrasound network for a global analysis of thermally induced atmospheric tides"

_Atmospheric Measurement Techniques, 2017_

## Referee Comment (RC1) · Anonymous Referee #1 · 19 Jan 2018

Infrasound research has seen a significant resurgence after the opening for signature in 1996 of the Comprehensive nuclear Test Ban Treaty (CTBT). The expansion of the global infrasound International Monitoring System (IMS) and experiment arrays deployed worldwide offers an opportunity to study background noise on a global scale. During the last decade, infrasound has developed into a broad interdisciplinary fields (combining acoustic and seismic technologies, atmospheric science, meteorology and climate) reinvigorating research field in geophysics.

Sparse observations in the middle atmosphere limit the ability to faithfully reproduce the dynamics of the middle atmosphere in numerical weather prediction and climate mod-

els. Recent studies have shown the capability of the IMS infrasound network to measure atmospheric perturbations in a wide frequency range, far beyond the frequency band of interest to detect explosions. In particular, the majority of IMS arrays measure pressure fluctuations with frequencies ranging from DC to tens of Hertz, encompassing gravity wave domain and atmospheric tides. Such network provides independent ground-based measurements useful for Numerical Weather Prediction models. The pressure measurements continuously recorded by the worldwide IMS infrasound network therefore constitute a unique set of data that could improve our knowledge on large scale atmospheric perturbations which impact the atmospheric dynamics.

The paper by P. Hupe et al. "Using barometric time series of the IMS infrasound network for a global analysis of thermally induced atmospheric tides" [Paper #amt-2017-465] demonstrates the potential of the IMS infrasound network to better characterize thermally induced atmospheric tides. This paper investigates the seasonal and latitudinal variability of atmospheric tides using historical recordings of the IMS arrays composed of high-quality microbarometers. The temporal and spectral methods presented here allow detailed geographic and temporal variability analyses of the solar and lunar tides' representation.

Compared with MERRA-2 products, these new observations provide additional quantitative measures of solar tide harmonics as well as low-amplitude gravitationally forced lunar tides. Is is shown that continuous IMS records represent well the surface pressure fields of solar tides and allow quantifying their seasonal and latitudinal fluctuations. This multi-year and global dataset open doors for further investigations into the source of atmospheric tides and their interaction with planetary waves or gravity waves. Given the accuracy and high temporal resolution of the barometric data from the IMS network, such observations are of high value for continuously calibrating sensors in a wide frequency band using the ambient infrasound noise, refining the knowledge of atmospheric dynamics and data assimilation problems.

Following Marty et al. (2010), this is the first systematic and global study of contin-

uous absolute pressure recordings. The representation of thermal tides in Numerical Weather Prediction models, often limited to the diurnal and the semidiurnal tides, would benefit from this high-resolution and uniform global observations.

Considering the new materials presented in this paper, the analyses carried out, and its revisions compared with an precedent submitted version, I recommend that this paper should be published in Atmospheric Measurement Techniques, as is.

PS: I would only suggest to add a time scale on Figure 2 (from January?). It seems that the annual variation is not cyclical. Why?

Please also note the supplement to this comment: https://www.atmos-meas-tech-discuss.net/amt-2017-465/amt-2017-465-RC1-supplement.pdf

---

## Referee Comment (RC2) · Anonymous Referee #2 · 19 Jan 2018

This paper provides a nice analysis of barometric data from the International Monitoring System infrasound network as a means to explore atmospheric tides. The paper is well written and the analysis, and comparisons with MERRA, provide an interesting and new set of observations.

My only suggestion to the authors would be to provide some more detail on the novelty of this dataset in comparison with other barometric datasets available. As an infrasound specialist, I am aware of the uniqueness of the microbarometer recordings, but the novelty of the absolute pressure recordings in comparison to what other datasets exists is not immediately apparent to me. Perhaps more discussion on other datasets

that provide this data would be useful.

---

## Author Comment (AC2) · 8 Mar 2018

Dear referee,

thank you for taking your time for the positive review. We have added a short discussion on other (global) barometric data sets (including additional references) in section 2 of our paper. In the added paragraph, we refer to the Global Observing System of the WMO and point out the beneficial features of the IMS infrasound network for studying atmospheric dynamics.

A marked-up manuscript version is uploaded as a supplement to this response.

[Figure]

Please also note the supplement to this comment:
https://www.atmos-meas-tech-discuss.net/amt-2017-465/amt-2017-465-AC2-
supplement.zip

---

## Author Response (AR1)

**Point-by-point response to the reviews of the manuscript #amt-2017-465 (Hupe et al.)**

**Referee #1 comment**

5    Infrasound research has seen a significant resurgence after the opening for signature in 1996 of the Comprehensive nuclear Test Ban Treaty (CTBT). The expansion of the global infrasound International Monitoring System (IMS) and experiment arrays deployed worldwide offers an opportunity to study background noise on a global scale. During the last decade, infrasound has developed into a broad interdisciplinary fields (combining acoustic and seismic technologies, atmospheric science, meteorology and climate) reinvigorating research field in geophysics.

10   Sparse observations in the middle atmosphere limit the ability to faithfully reproduce the dynamics of the middle atmosphere in numerical weather prediction and climate models. Recent studies have shown the capability of the IMS infrasound network to measure atmospheric perturbations in a wide frequency range, far beyond the frequency band of interest to detect explosions. In particular, the majority of IMS arrays measure pressure fluctuations with frequencies ranging from DC to tens of Hertz, encompassing gravity wave domain and atmospheric tides. Such network provides independent ground-based measurements

15   useful for Numerical Weather Prediction models. The pressure measurements continuously recorded by the worldwide IMS infrasound network therefore constitute a unique set of data that could improve our knowledge on large scale atmospheric perturbations which impact the atmospheric dynamics.

The paper by P. Hupe et al. "Using barometric time series of the IMS infrasound network for a global analysis of thermally induced atmospheric tides" [Paper #amt-2017-465] demonstrates the potential of the IMS infrasound network to better

20   characterize thermally induced atmospheric tides. This paper investigates the seasonal and latitudinal variability of atmospheric tides using historical recordings of the IMS arrays composed of high-quality microbarometers. The temporal and spectral methods presented here allow detailed geographic and temporal variability analyses of the solar and lunar tides' representation. Compared with MERRA-2 products, these new observations provide additional quantitative measures of solar tide harmonics as well as low-amplitude gravitationally forced lunar tides. It is shown that continuous IMS records represent well the surface

25   pressure fields of solar tides and allow quantifying their seasonal and latitudinal fluctuations. This multi-year and global dataset open doors for further investigations into the source of atmospheric tides and their interaction with planetary waves or gravity waves. Given the accuracy and high temporal resolution of the barometric data from the IMS network, such observations are of high value for continuously calibrating sensors in a wide frequency band using the ambient infrasound noise, refining the knowledge of atmospheric dynamics and data assimilation problems.

Following Marty et al. (2010), this is the first systematic and global study of continuous absolute pressure recordings. The representation of thermal tides in Numerical Weather Prediction models, often limited to the diurnal and the semidiurnal tides, would benefit from this high-resolution and uniform global observations.

Considering the new materials presented in this paper, the analyses carried out, and its revisions compared with a precedent submitted version, I recommend that this paper should be published in Atmospheric Measurement Techniques, as is.

PS: I would suggest to add a time scale on Figure 2 (from January?). It seems that the annual variation is not cyclical. Why?

**Authors' response to Referee #1**

Dear referee,

we highly appreciate your comprehensive and positive review. Given the schematic character of Figure 2, we initially assumed that a detailed time scale would not be necessary. We have now added the time scale (Jan-Dec). For reasons of plausibility, we have also chosen a data set that is cyclical. Therefore, Figure 2 has been changed in the revised manuscript (see below). Further changes can be found in the marked-up manuscript version (supplement to our response to Referee #2).

Changes in the manuscript:

[Figure]

Figure 2: Schematic illustration for handling an annual data set of a four-sensor array. Different pressure at the various sensors (a) can result from the calibration or from different altitudes since the arrays have an aperture of 1-3 km (Evers and Haak 2010). Here, temporarily missing data for sensors 2 and 4 are without significant consequence when deriving the median (b) as time series for a multi-sensor station.

**Referee #2 comment**

This paper provides a nice analysis of barometric data from the International Monitoring System infrasound network as a means to explore atmospheric tides. The paper is well written and the analysis, and comparisons with MERRA, provide an interesting and new set of observations.

My only suggestion to the authors would be to provide some more detail on the novelty of this dataset in comparison with other barometric datasets available. As an infrasound specialist, I am aware of the uniqueness of the microbarometer recordings, but the novelty of the absolute pressure recordings in comparison to what other datasets exists is not immediately apparent to me. Perhaps more discussion on other datasets that provide this data would be useful.

**Authors' response to Referee #2**

Dear referee,

thank you for taking your time for the positive review. We have added a short discussion on other (global) barometric data sets (including additional references) in section 2 of our paper. In the added paragraph, we refer to the Global Observing System of the WMO and point out the beneficial features of the IMS infrasound network for studying atmospheric dynamics.

Changes in the manuscript:

*A marked-up manuscript version is uploaded as a supplement to the response (see below).*

**List of all relevant changes made in the manuscript**

- Section 2: We have added a paragraph on other (global) barometric data sets to point out the added value of the IMS infrasound stations' barometric data; other paragraphs in Section 2 have been adapted accordingly (response to Referee #2).
- Figure 2: We have added a time scale of the schematic illustration and changed the data basis; the caption has been adapted accordingly (response to Referee #1).
- References: We have added five entries, corresponding to the changes made in Section 2.

**Marked-up version of the manuscript**

Please see below.

[revised manuscript text omitted]